# Aerodynamics Analysis of Helicopter Rotor in Flight Test Using Strain Gauge Sensors

**DOI:** 10.3390/s25061911

**Published:** 2025-03-19

**Authors:** Shuaike Jiao, Jiahong Zheng

**Affiliations:** 1Chinese Flight Test Establishment, Xi’an 710089, China; jsk1220955@163.com; 2Applied Mechanics Lab, Center for Nano and Micro Mechanics and State Key Laboratory of Flexible Electronics Technology, Department of Engineering Mechanics, Tsinghua University, Beijing 100084, China

**Keywords:** strain gauge sensors, flight test, helicopter rotor, flapping moment, equivalent aerodynamic force

## Abstract

The acquisition of aerodynamic loads on helicopter rotors is fundamental to the study of helicopter performance optimization, structural design, flight control, and other aspects. However, at present, aerodynamic loads on rotors are primarily obtained through theoretical calculations, simulation analysis, and wind tunnel tests, with few reports on flight measurements. This paper proposes a method for obtaining helicopter rotor aerodynamic loads by flapping moment measurements in flight with strain gauge sensors. First, strain gauge sensors are installed at different cross-sectional positions on the rotor blades to measure strain during flight. Then, the strains are incorporated into the blade flapping motion equations to establish the relationship between rotor aerodynamic loads and flapping moment. Finally, the aerodynamic loads on the rotor are calculated by the relationship. This method can provide more accurate load calculation results compared to simulation computations and wind tunnel tests. In this paper, the distribution patterns of rotor aerodynamic loads were investigated, which aligned with theoretical analysis and can offer valuable insights for blade design optimization.

## 1. Introduction

Helicopters are now applied in various aspects of human life. In the civilian sector, they are used for large-scale construction hoisting operations, offshore oil exploration, forest fire suppression, and more. In the military domain, they play roles in counter-terrorism, military exercises, and personnel transportation. It is evident that, both in civilian and military applications, helicopters hold significant importance. However, compared to fixed-wing aircraft technology, helicopter technology is not yet as mature, with a common issue being their relatively short lifespan. The rotor is one of the core components of a helicopter, which bears periodic alternating load in centrifugal field and flow field [1]. The aerodynamic load, which is the main component of fatigue load, is more difficult to solve than centrifugal force. Therefore, the research being conducted on helicopter aerodynamics is essential [2,3], and how to determine the distribution law of aerodynamics is particularly important.

Scholars both domestically and internationally have conducted research on the identification and calculation of aerodynamic loads on helicopter rotor blades [4,5]. Daughaday and Kline [6] used measured blade bending moments to identify generalized aerodynamic forces based on a single blade mode. Loewy [7] identified generalized aerodynamic forces using measured blade bending moments through third-order modes. Scheiman [8] measured the flapping bending moments and surface aerodynamic pressures of the rotor blades of a CH-33 helicopter. DuWaldt et al. [9] discovered that the phase of the blade was the reason why the calculation of generalized aerodynamic loads using measured sectional bending moments was not ideal. Bartlett F.D and Flannelly W.D. identified the dynamic forces of the helicopter rotor shaft based on the frequency domain method and determined the main harmonic frequency components transmitted through the hub center [10]. N. Giansante, R. Jones, and N. J. Calapodas identified the external forces on the main and tail rotors of an AH-1G helicopter during flight by directly inverting the frequency response function matrix based on measured acceleration responses and the structural system’s frequency response function matrix [11]. Zhang Jinghui et al. used the strain transfer function matrix method to identify the six force components at the rotor hub on the main rotor shaft [12]. William G. Bousman established a method to identify aerodynamic loads in the flapping direction using measured flapping bending moments [13], known as the modal analysis method. In the same year, Liu Shoushen and Davies G.A.O. identified the aerodynamic loads on the blade using measured strains [14]. Liu Shoushen proposed a method combining theory and experiments to identify rotor blade loads [15], which involved converting measured rotor blade bending moments to derive aerodynamic loads. Liu Shoushen, Jerry P. Higman, and Daniel P. Schrage [16] detailed the methods for identifying rotor blade loads, load conversion, and hub load identification, providing a new theoretical foundation and methods for subsequent research. Liu Shoushen, Jerry P. Higman, and Daniel P. Schrage established the “inverse transfer matrix method” for blade load identification under a coupled flapping-lag-torsion rotor dynamics model [17]. Jerry P. Higman, Liu Shoushen, and Daniel P. Schrage, based on Jerry P. Higman’s doctoral research, introduced methods for inflow and load identification for flap–lag–torsion coupled rotor blades [18], extending blade load identification to more complex models and improving accuracy. Jerry P. Higman published a method for identifying torsional aerodynamic loads in Japan [19]. In 2000, Liu Shoushen, Daniel P. Schrage, and Jerry P. Higman published several new methods to improve the accuracy of rotor blade load identification [20], including composite modal analysis, force analysis, and composite force-modal analysis, each addressing issues encountered in previous load identification processes. Liu Shoushen, Daniel P. Schrage, and Jerry P. Stephen [21] used strain sensors to measure blade strain during helicopter ground runs and transmitted the data to a ground station via battery-powered wireless telemetry. Higman [22] summarized helicopter rotor blade load identification methods and detailed several latest methods in the field, including rotor inflow identification and load conversion methods. In 2008, Chen Wen et al. established a new method for identifying distributed loads on rotor blades based on fiber optic sensing technology, successfully identifying radially distributed loads on model rotor blades [23]. Wu Chao et al. studied the identification of rotor vibration loads based on the linear superposition assumption of modal bending moments [24]. Simone Weber and Thomas Kissinger [25] published an article in 2021 on the application of fiber optic sensing systems in blade structural dynamics, comparing two types of fiber optic sensors for measuring load characteristics. Zhang Honglin et al. identified random and periodic vibration loads on helicopters using the inverse virtual excitation method [26]. Li Zheng et al. [27] modeled the distributed dynamic loads on rotor blades as continuously distributed periodic loads, fitted them using generalized orthogonal basis functions, and used finite element analysis software to calculate the dynamic response of the rotating blade model under orthogonal basis function excitation, achieving dynamic calibration for distributed dynamic load identification. In the same year, Chen Guangjiong [28] identified distributed dynamic loads on helicopter rotor blade models based on the frequency domain method. Zhang Honglin [29] obtained the structural dynamic loads at different sectional positions of the blade using fiber Bragg grating sensors.

To date, while numerous studies have been conducted both domestically and internationally on the aerodynamic identification and calculation of helicopter rotors, most reported methods are based on simulation experiments, laboratory environments, or wind tunnel testing environments, with few reports on actual in-flight measurements.

In this paper, a method for obtaining helicopter rotor aerodynamic loads by strain gauge sensors is proposed. First, strain gauge sensors are installed at different cross-sectional positions on the rotor blades to measure strain during flight. Optical strain measurement techniques, such as Digital Image Correlation (DIC) [30], can obtain the strain field of the entire blade. However, optical measurement techniques impose high demands on the performance and installation location of the optical measurement equipment, and also require a high level of cleanliness on the blade surface. In contrast, the electrical resistance strain gauge method is a mature technology. Therefore, this paper adopts electrical resistance strain gauges to measure the strain of the blade. Then, the strains are incorporated into the blade flapping motion equations to establish the relationship between rotor aerodynamic loads and flapping moment. Finally, the aerodynamic loads on the rotor are calculated by the relationship. Compared with simulation and wind tunnel testing, this method is based on the measured strain data of the flight test with strain gauge sensors, and therefore, the calculation results are more accurate. In addition, this method is simple and more valuable for engineering applications.

## 2. Flapping Motion Model with Strain

During the flight of a helicopter, the rotor blades undergo flapping motion. The theoretical derivation is based on the theory of elastic beams, which approximates the rotor blade as a cantilever beam and assumes that the blade is not completely rigid but has some deformation. A schematic diagram of the rotor blade is shown in Figure 1, where ① is the “flapping hinge”, which is designed to prevent excessive bending moments at the blade root; ② is the rotor blade; and ③ is the rotor disk formed when the blade is in operation.

During operation, the blade undergoes periodic flapping motion, and it is assumed that the rotor speed remains constant. A force analysis on a single blade is shown in Figure 2. Here, *q*(*r*,*φ*) represents the aerodynamic forces acting on the blade, *q_G_* is the gravitational force of the blade, *q_L_*(*r*)is the centrifugal force generated by the high-speed rotation of the blade, and *q_β_*(*r*,*φ*) is the inertial force resulting from the up-and-down flapping motion of the blade. The flapping angle *β* is the angle between the rotating structural plane *S—S* and the blade, as shown in Figure 2a. Figure 2b provides a detailed view of the blade at an arbitrary flapping position, where the length of the blade is *R*.

The sum of moments generated by all forces on any section *r—r* of the blade can be expressed by Equation (1):(1)$$M_{q}+M_{G}+M_{L}+M_{\beta }=M_{C}$$
where $M_{q}$ refers to the moment of aerodynamic force; $M_{G}$ refers to the moment of gravity; $M_{L}$ refers to the moment of centrifugal force; $M_{\beta }$ refers to the moment of inertia force; and $M_{C}$ refers to the ***r—r*** section moment, which is obtained by flight test.

The moment of gravity is ignored because of its value is much smaller than others. Then, Equation (1) can be expressed as Equation (2).(2)$$M_{q}+M_{L}+M_{\beta }=M_{C}$$

The bending moment on the section ***r—r*** exerted by aerodynamic force at any position $r=r_{0}$ can be expressed as $q(r_{0},\varphi )(r_{0}-r)dr_{0}$, and total moment can be expressed as(3)$$M_{q}=\int_{r}^{R}q(r_{0},\varphi )(r_{0}-r)dr_{0}$$

The bending moment on the section ***r*—*r*** exerted by centrifugal force at any position $r=r_{0}$ can be expressed as $-q_{L}(r_{0})(r_{0}-r)\sin\beta dr_{0}$, and total moment can be expressed as(4)$$M_{L}=-\int_{r}^{R}q_{L}(r_{0})(r_{0}-r)\sin\beta dr_{0}$$

As the flapping angle is very small, sinβ≈β. The centrifugal force can be expressed as $q_{L}(r_{0})=mr_{0}\Omega^{2}$. Equation (5) can be obtained:(5)$$M_{L}=-\int_{r}^{R}m\Omega^{2}r_{0}(r_{0}-r)\beta dr_{0}$$

The bending moment on the section ***r*—*r*** exerted by inertia force at any position $r=r_{0}$ can be expressed as $-q_{\beta }(r_{0},\varphi )(r_{0}-r)dr_{0}$, and total moment can be expressed as follows:(6)$$M_{\beta }=-\int_{r}^{R}q_{\beta }(r_{0},\varphi )(r_{0}-r)dr_{0}$$

As $q_{\beta }(r_{0},\varphi )=ma=m\frac{d^{2}(\beta r_{0})}{dt^{2}}$ and $\frac{d^{2}\beta }{dt^{2}}=\frac{d^{2}\beta }{d\varphi^{2}}\cdot \frac{d^{2}\varphi }{dt^{2}}=\Omega^{2}\frac{d^{2}\beta }{d\varphi^{2}}$ are brought into Equation (6), we can obtain Equation (7):(7)$$M_{\beta }=-\int_{r}^{R}mr_{0}\Omega^{2}\frac{d^{2}\beta }{d\varphi^{2}}(r_{0}-r)dr_{0}$$

Equations (2), (3), (5) and (7) can be combined into a single equation as Equation (8):(8)$$\int_{r}^{R}q(r_{0},\varphi )(r_{0}-r)dr_{0}-\int_{r}^{R}m\Omega^{2}r_{0}(r_{0}-r)\beta dr_{0}-\int_{r}^{R}mr_{0}\Omega^{2}\frac{d^{2}\beta }{d\varphi^{2}}(r_{0}-r)dr_{0}=M_{C}$$

Equation (8) can be simplified as Equation (9):(9)$$\int_{r}^{R}q(r_{0},\varphi )(r_{0}-r)dr_{0}=m\Omega^{2}(\beta +\frac{d^{2}\beta }{d\varphi^{2}})(\frac{R^{3}}{3}-\frac{R^{2}}{2}r+\frac{1}{6}r^{3})+M_{C}$$

The flapping angle *β* can be written as a Fourier series:(10)$$\beta =a_{0}-a_{1}\cos\varphi -b_{1}\sin\varphi -a_{2}\cos2\varphi -b_{2}\sin2\varphi -$$
where $a_{0}$ is the coning angle of the helicopter’s rotor blade. Considering only the first harmonic, Equation (10) can be simplified as Equation (11):(11)$$\beta =a_{0}-a_{1}\cos\varphi -b_{1}\sin\varphi$$

The measured bending moment $M_{C}$ in the flight test is a function of *r* and *φ*. Assume that there is no coupling between *r* and *φ*.(12)$$M_{C}=M_{C1}(r)M_{C2}(\varphi )$$

Assume that the aerodynamic force is uniformly distributed between *r* and *R*.(13)$$q_{eq}=\frac{2\int_{r}^{R}q(r_{0},\varphi )(r_{0}-r)dr_{0}}{{\left(R-r\right)}^{2}}$$

Equations (9) and (11)–(13) can be combined into Equation (14):(14)$$\frac{q_{eq}{\left(R-r\right)}^{2}}{2}=M_{C1}(r)M_{C2}(\varphi )+m\Omega^{2}a_{0}(\frac{R^{3}}{3}-\frac{R^{2}}{2}r+\frac{1}{6}r^{3})$$
where $M_{C1}(r)=\sum_{i=1}^{n}a_{n}r^{n}$ and Equation (14) is normalized to Equation (15).(15)$$q{\left(r,\varphi \right)}_{eq}\cdot R^{2-n}=\frac{2M_{C1}(\bar{r})M_{C2}(\varphi )}{{\left(1-\bar{r}\right)}^{2}}+\frac{2m\Omega^{2}a_{0}}{{\left(1-\bar{r}\right)}^{2}}(\frac{1}{3R^{n-3}}-\frac{1}{2R^{n-3}}\bar{r}+\frac{1}{6R^{n-3}}{\bar{r}}^{3})$$
where $0\leq \bar{r}=\frac{r}{R}\leq 1$.

## 3. The Flight Measurement of Blade Strains

$M_{C1}(r)$ and $M_{C2}(\varphi )$ were obtained with strain gauge sensors by a flight test on a helicopter. According to Figure 3, the locations of strain gauge sensors mounted on blade are $\bar{r}=0,0.2,0.4,0.6,0.8$. Resistance strain gauges are installed at the quarter-chord positions (the pitch axis of the blade) on the upper and lower surfaces of each measurement section of the blade and are connected to form a full-bridge circuit, which is used to measure the bending moment of the blade in the flapping direction. The blade is made of composite materials, and the resistance strain gauges are installed on the surface of the blade using an adhesive. The technical parameters of the sensor are presented in Table 1.

The load calibration test in the ground is carried out to obtain the relationship between the flapping moment and flapping strain. According to Figure 4, different loads were hung on the free end of blade to produce gradient strains. The strain data were recorded by the KAM500 data collector. The load calibration curves are seen in Figure 5 and Table 2. The load calibration curves are linear. Figure 5 shows the linear equations between the bending moment and the strain of each measurement section of the blade obtained through ground calibration. During the flight measurement test, the data collector records the strain from the strain gauges at each measurement section in real time, and then the flight bending moment on the blade can be deduced.

The moment measurement technology of the blade section belongs to dynamic load measurement technology in the flight test. The equipment for recording load data is called rotor collector, which is fixed above hub, and it rotates synchronously with the rotor hub during flight. The test channels on the moving parts such as blades, rotor shaft and pitch link are connected with the channels on the collector. Then, the data are transmitted to the cabin and the data are unloaded after the flight. Through helicopter flight measurements, the collector records strain data in real time. The blade load in the flight state can be obtained by using the ground calibration equation for the load. The flight test flow chart is shown in Figure 6. The installation position of collector is shown in Figure 7, where I is the platform of main retarder, II is the main retarder, III is the rotor shaft, IV is the hub, V is the disk of the helicopter, and VI is the collector. The actual testing scene can be seen in Figure 8.

Assuming that $M_{C1}(\bar{r})$ and $M_{C2}(\varphi )$ are mutually independent, the distribution of $M_{C1}(\bar{r})$ can be determined when the azimuth angle is fixed. The scatter plots for different azimuth angles are shown in Figure 9. By fitting the scatter plots, the fitting curves are obtained, as shown in Figure 10. The accuracy of the fitting curves was analyzed, and the residual plots for each point are shown in Figure 11.

From Figure 10, it can be observed that the spanwise distribution patterns of each section are generally consistent. After approximating and fitting using a polynomial, the following Equation (16) is obtained.(16)$$M_{C1}(\bar{r})=0.017+3.431\bar{r}-0.730{\bar{r}}^{2}-7.627{\bar{r}}^{3}+4.840{\bar{r}}^{4}+3.334{\bar{r}}^{5}-8.772{\bar{r}}^{6}+0.539{\bar{r}}^{7}+4.672{\bar{r}}^{8}$$

The variance of the fitting value is 0.98385 and the residual value is small, so the fitting result is credible.

The measured moment of the helicopter when flying stably at a certain speed is shown in Figure 12.

The first 10 s of the data is transformed by Fourier transform to obtain the spectrum, as shown in Figure 13. It can be found from Figure 13 that the frequency distribution of the load is mainly concentrated in the first four frequencies. Therefore, the amplitude and phase of the first four frequencies are selected for fitting. According to Figure 14, the fitting result is essentially consistent with the measured load value of the flight test.

## 4. Calculation of Aerodynamics

Based on the moment distribution pattern presented in Section 3, the moment values and equivalent aerodynamic load values of the rotor under different helicopter flight conditions were analyzed. Figure 13 shows the full-field moment distribution pattern of the helicopter under steady level flight conditions for various airspeeds. Figure 15a, Figure 15b, Figure 15c, Figure 15d, Figure 15e, and Figure 15f correspond to the moment distribution patterns of the rotor disk at airspeeds of 80 km/h, 100 km/h, 120 km/h, 140 km/h, 160 km/h, and 180 km/h, respectively.

The moment distribution of the blade at different level flight speeds and the same azimuth is shown in Figure 16. According to Figure 16, it can be found that the maximum moment section is about *r* = 0.4. The calculation results can be used to determine the installation position of strain sensors during the helicopter flight test.

The load data when the helicopter flies stably at different indicated airspeeds are brought into the model to calculate the aerodynamic forces, as shown in Figure 17.

The aerodynamic forces distribution of the blade at different level flight speeds and the same azimuth is shown in Figure 18. According to Figure 18, it can be found that the aerodynamic force increases first and then decreases from the blade tip to blade root and the maximum aerodynamic force section is about *r* = 0.7. The calculation results are consistent with the literature [25], which shows the model is accurate.

The aerodynamic statistical analysis shows that the aerodynamic force presents a negative exponential distribution, as shown in Figure 19. The aerodynamic force is dominated by small load, which conforms to the original intention of blade design. The statistical results can be used for structural health monitoring of rotors.

## 5. Conclusions

In this paper, a flapping motion model of blades which are based on the elastic beam theory is established. The flapping moment and aerodynamic load are connected through the flapping motion model, and the aerodynamic load is directly solved through the measured flapping moment of a flight test with strain gauges sensors. A concept of “equivalent aerodynamic force” is proposed in this paper, and it is assumed that the radial and circumferential flapping moments are independent of each other. Then, the load data measured by a flight test is brought into the model for calculation and analysis, and the following four conclusions are obtained:(1)The radial distribution of flapping moment increases first and then decreases, which is related to the existence of flapping hinge, and the maximum moment load is about *r* = 0.4;(2)The radial distribution of aerodynamic force increases first and then decreases, which is related to the existence of the flow field of blades, and the maximum aerodynamic force is about *r* = 0.7 which is consistent with the calculation results in the literature;(3)The aerodynamic force of the helicopter rotor presents a negative exponential distribution, and the values are concentrated between 0 and 0.4;(4)The main component of aerodynamic force is smaller load, which accounts for nearly 90%.

In this paper, the calculation results can be used as a basis to select the installation position of the sensor, verify and improve the helicopter blade design, and optimize the blade structure.

## Figures and Tables

**Figure 1 sensors-25-01911-f001:**
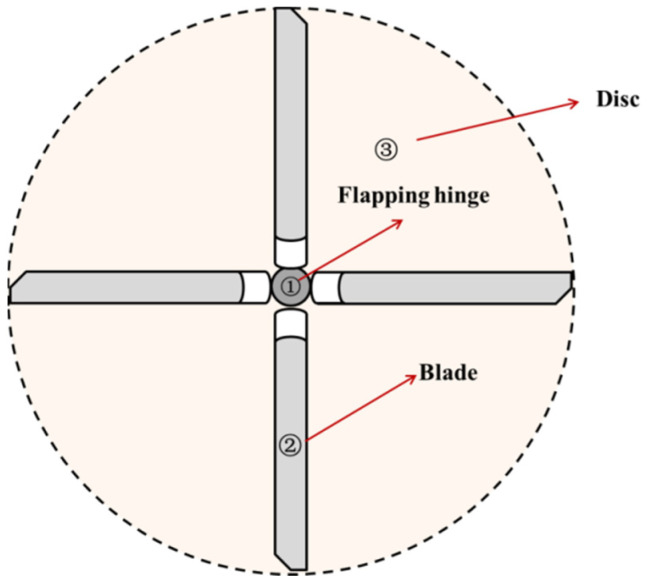
Rotor structure.

**Figure 2 sensors-25-01911-f002:**
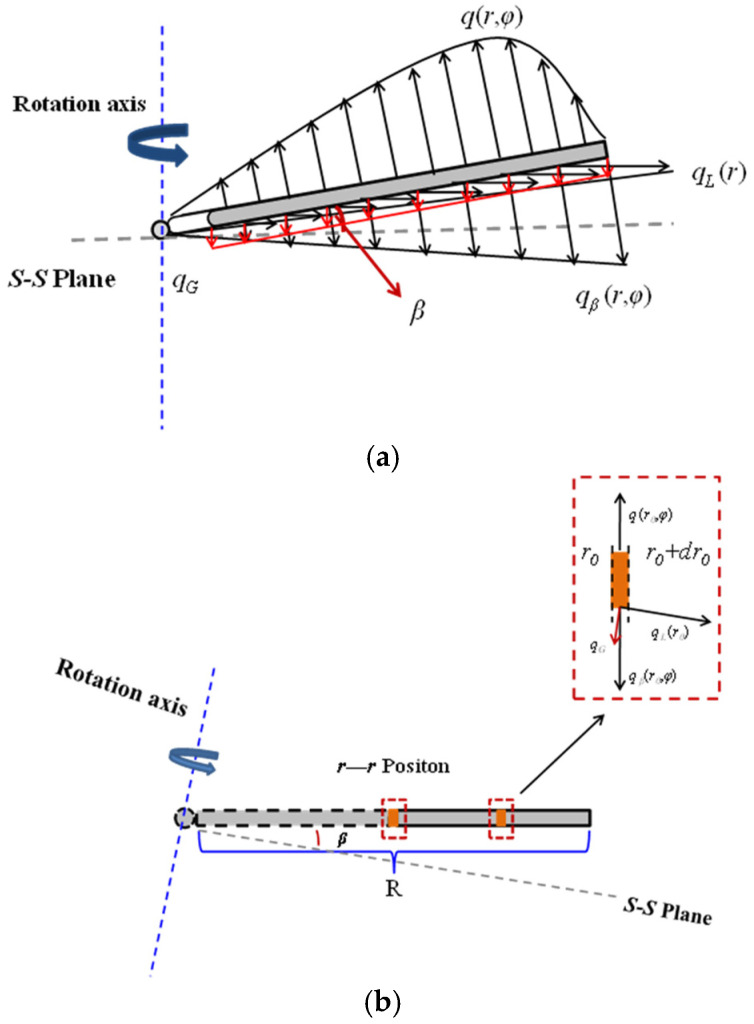
Blade force analysis. (**a**) Force diagram of the single blade. (**b**) Stress analysis diagram of any section r—r of the blade.

**Figure 3 sensors-25-01911-f003:**
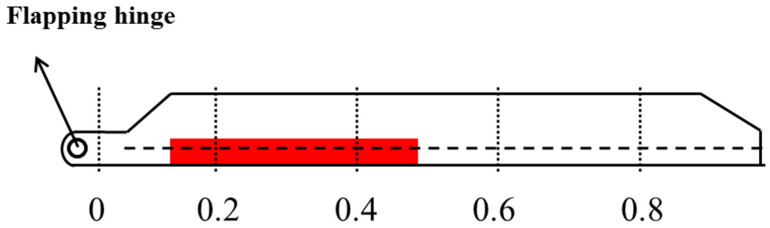
Locations of strain gauge sensors mounted on blade.

**Figure 4 sensors-25-01911-f004:**
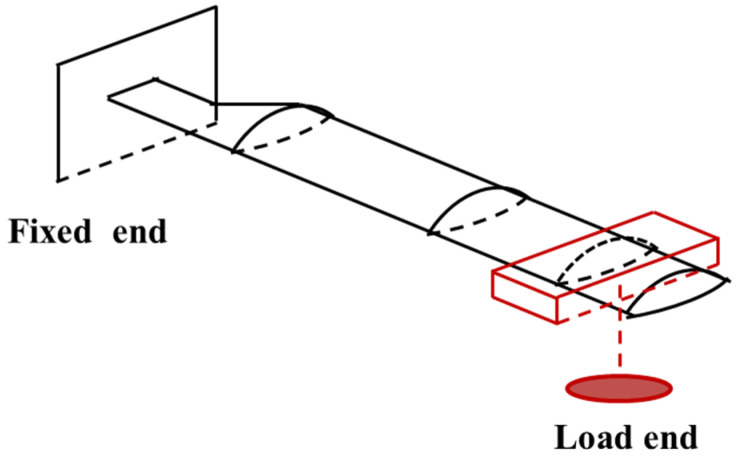
Load calibration test.

**Figure 5 sensors-25-01911-f005:**
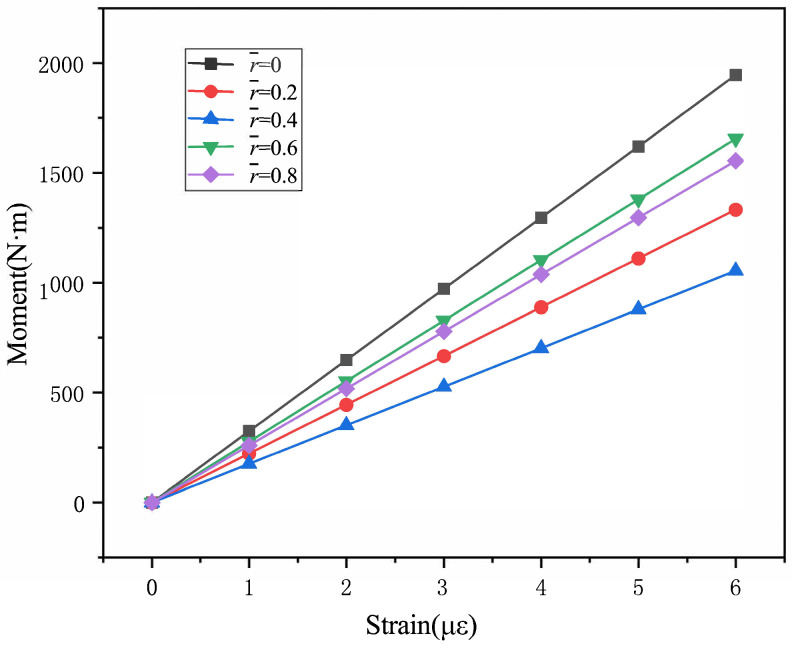
The load calibration curves of ground test.

**Figure 6 sensors-25-01911-f006:**
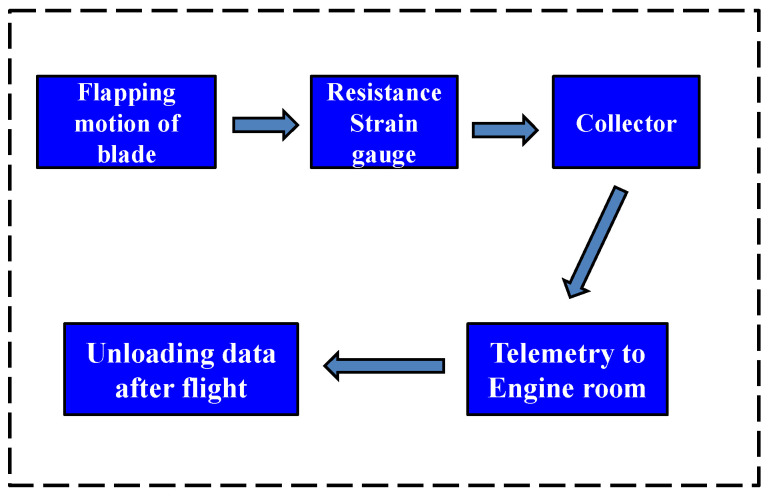
The flight test flow chart.

**Figure 7 sensors-25-01911-f007:**
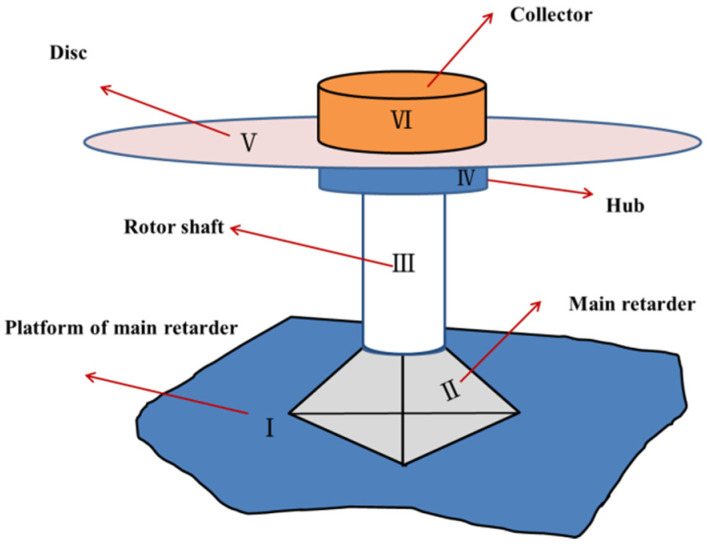
Installation position of collector.

**Figure 8 sensors-25-01911-f008:**
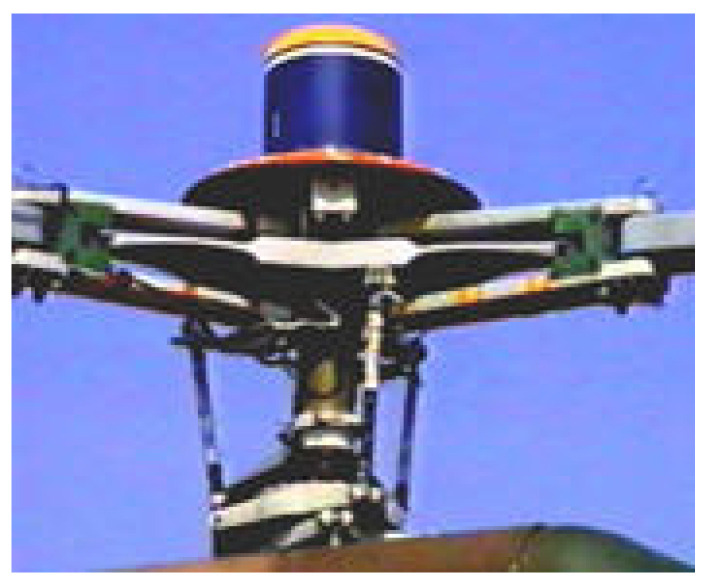
The actual testing scene.

**Figure 9 sensors-25-01911-f009:**
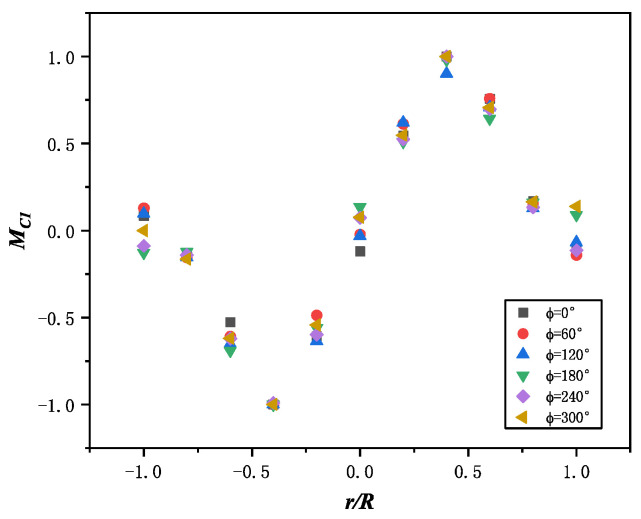
Moment of every section in different azimuth.

**Figure 10 sensors-25-01911-f010:**
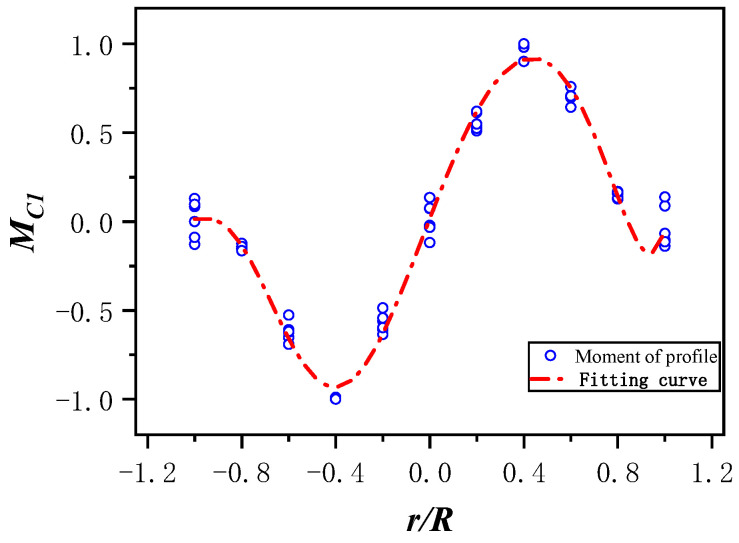
Scatter fitting.

**Figure 11 sensors-25-01911-f011:**
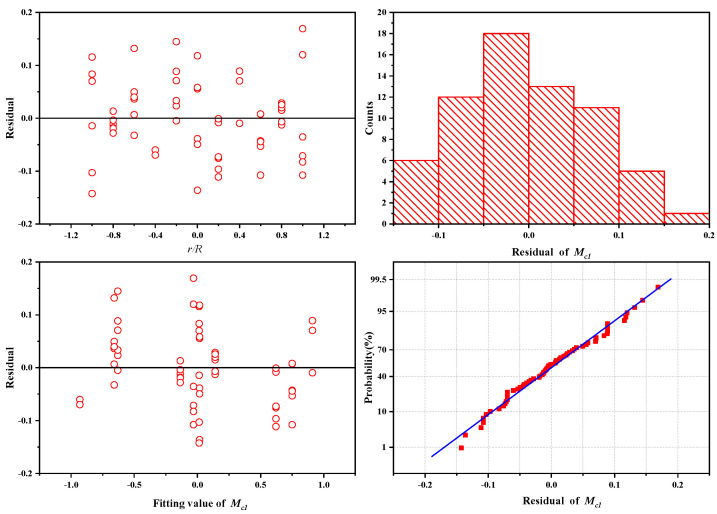
Residual plots of the fitting curve.

**Figure 12 sensors-25-01911-f012:**
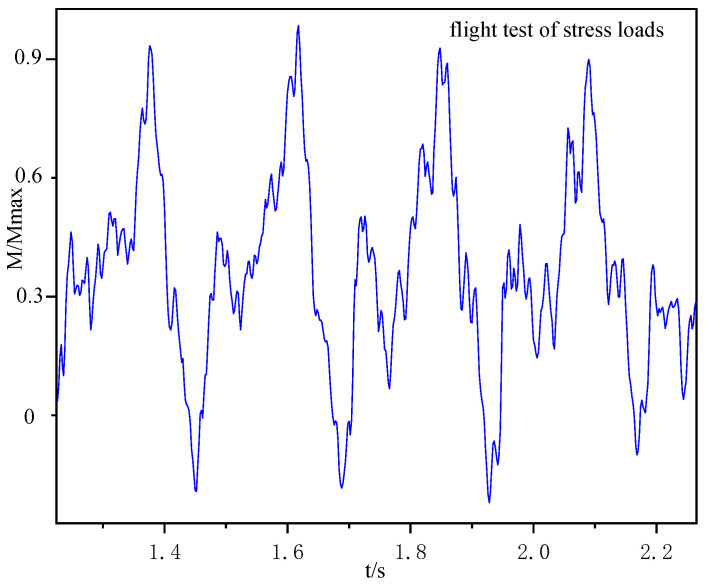
Moment of helicopter in level flight.

**Figure 13 sensors-25-01911-f013:**
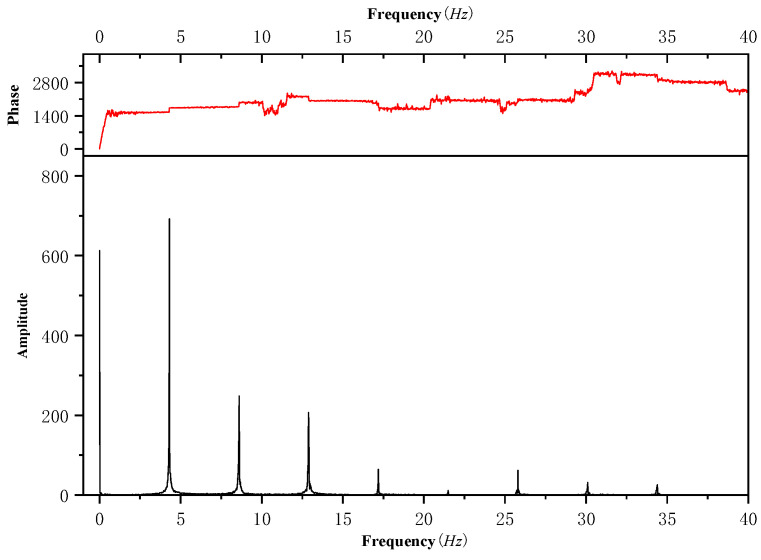
Frequency spectrum.

**Figure 14 sensors-25-01911-f014:**
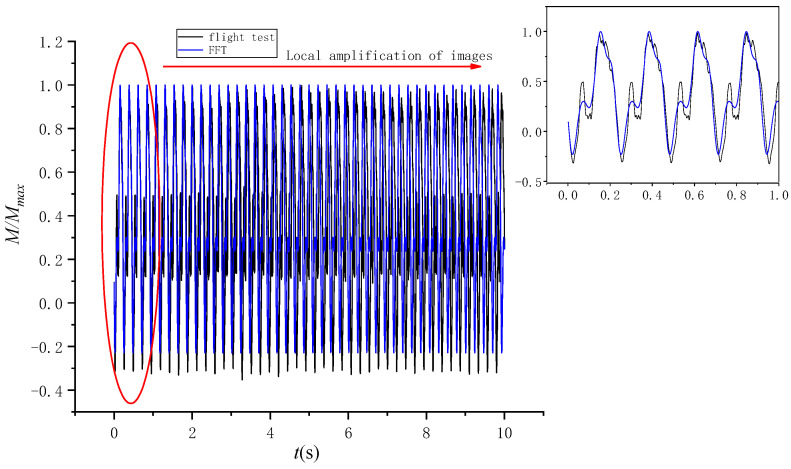
Fourier fitting curve.

**Figure 15 sensors-25-01911-f015:**
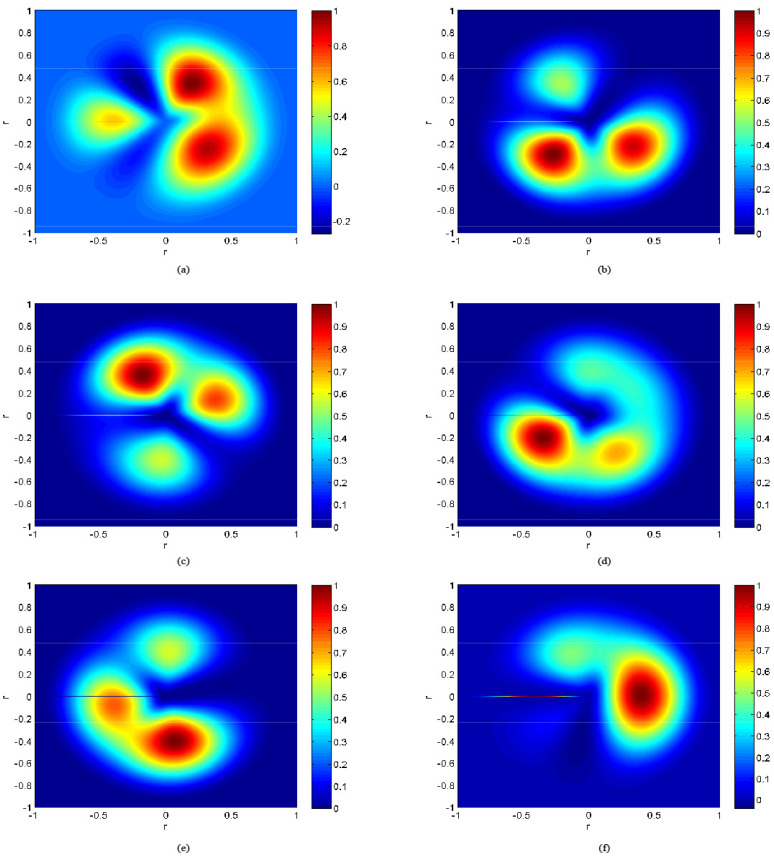
Moment of the helicopter at indicated airspeeds. (**a**) indicated airspeeds *V_i_* = 80 km/h. (**b**) indicated airspeeds *V_i_* = 100 km/h. (**c**) indicated airspeeds *V_i_* = 120 km/h. (**d**) indicated airspeeds *V_i_* =140 km/h. (**e**) indicated airspeeds *V_i_* = 160 km/h. (**f**) indicated airspeeds *V_i_* = 180 km/h.

**Figure 16 sensors-25-01911-f016:**
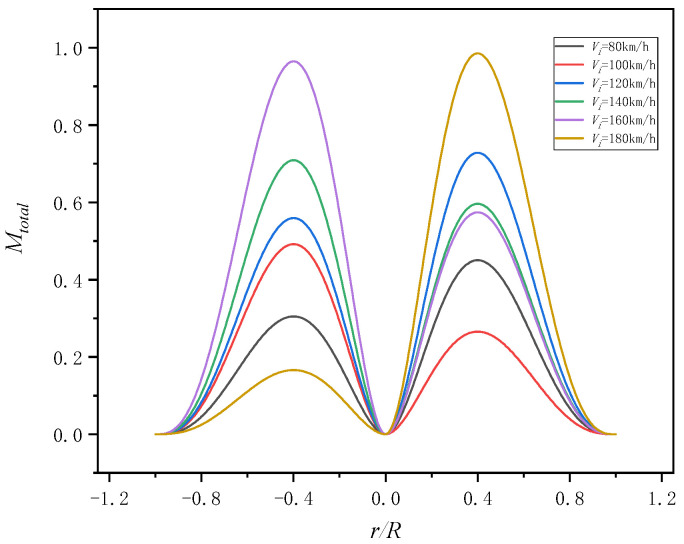
Radial distribution of moment.

**Figure 17 sensors-25-01911-f017:**
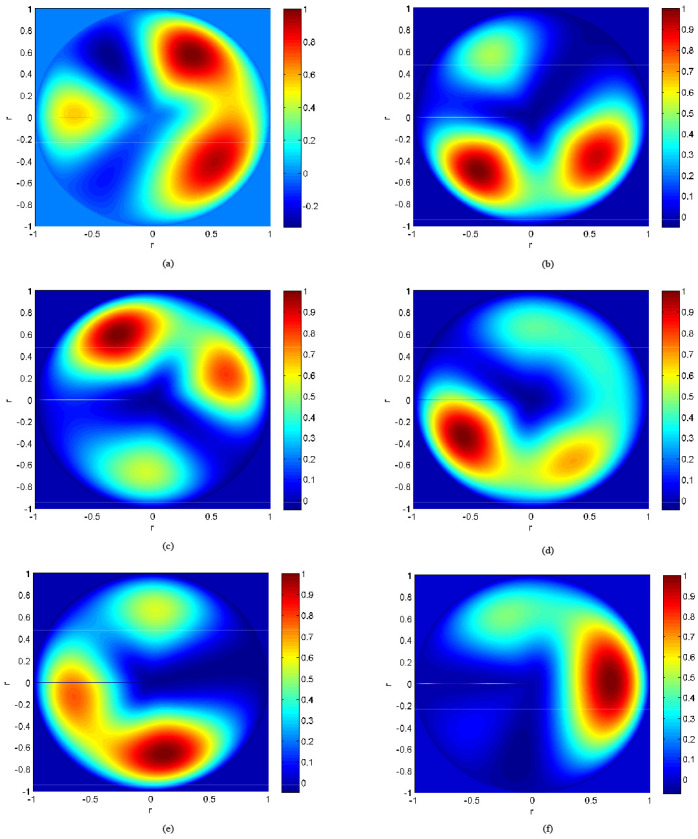
Aerodynamic force of the helicopter at indicated airspeeds. (**a**) indicated airspeeds *V_i_* = 80 km/h. (**b**) indicated airspeeds *V_i_* = 100 km/h. (**c**) indicated airspeeds *V_i_* = 120 km/h. (**d**) indicated airspeeds *V_i_* = 140 km/h. (**e**) indicated airspeeds *V_i_* = 160 km/h. (**f**) indicated airspeeds *V_i_* = 180 km/h.

**Figure 18 sensors-25-01911-f018:**
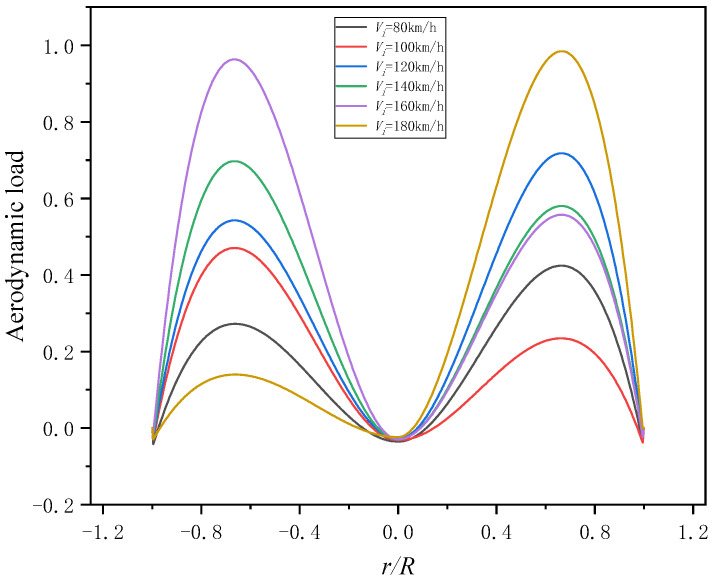
Radial distribution of aerodynamic forces.

**Figure 19 sensors-25-01911-f019:**
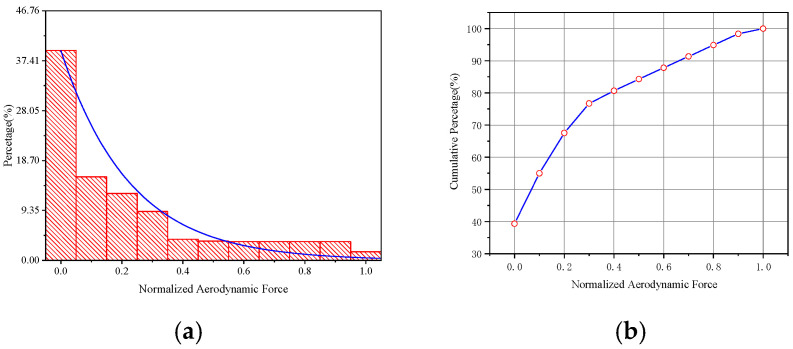
Aerodynamic statistical analysis. (**a**) Statistical distribution of aerodynamic loads (**b**) Cumulative percentage of the proportion of aerodynamic loads.

**Table 1 sensors-25-01911-t001:** Technical parameters of the resistance strain gauge sensor.

Type	Resistance/(Ω)	Sensitivity Coefficient
Resistance Strain Gauge BA350-2AA(9)-G150-JQC	349.9 ± 0.2	2.12 ± 0.01

**Table 2 sensors-25-01911-t002:** The load calibration curves of ground test.

Section	Curves (*x* (mV/V), *y* (N·m))
0	y = 324.22x − 0.1107
0.2	y = 222.14x − 0.1302
0.4	y = 175.72x − 0.1332
0.6	y = 276.07x − 0.0222
0.8	y = 259.3x + 0.0322

## Data Availability

Data are contained within the article.

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
