# Peer review of "Aerodynamics Analysis of Helicopter Rotor in Flight Test Using Strain Gauge Sensors"

_sensors, 2025, doi:10.3390/s25061911_

Round 1

Reviewer 1 Report

Comments and Suggestions for Authors

This paper proposes a method for obtaining helicopter rotor aerodynamic loads by flapping moment measurements in flight with strain gauge sensors. The results are acceptable. Some suggestions are given as follows:

(1) please give the application of Eqs. 1-15 in the paper.

(2) please provide the figure of the actual testing scene.

Author Response

 Firstly, we appreciate your helpful comments and suggestions. We have modified the manuscript carefully. The attachment is the list of our itemized answers and changes.

Reviewer 2 Report

Comments and Suggestions for Authors

The paper is devoted to the acquisition of aerodynamic loads, particularly flapping moment acting on helicopter rotor blade. The main feature of the study is the research data acquisition during flight measurements instead of common simulation results and lab testings. The topic of research seems relevant but at the same time literature review looks a bit outdated

Comments:

  • When explaining loads acting upon rotor blade in lines 119-127 authors use x, and φ as coordinates, but in Figure 2, r and φ is used. The designations should be the same throughout the article.

  • Text in lines 144 and 148 is the same despite the fact that in line 148 the moment of centrifugal force is explained

  • What is Ω in the equation of centrifugal force?

  • line 174 “The Eqs. (14) can be derived by the Eqs. (9), (11), (12) and (13)”. There is no equation (9)

  • On Figure 3 some red rectangle is shown. It is not clear what it represents.

  • Despite the fact that the locations of strain gauges are shown it is not clear what orientations they have which is important in understanding the direction of strain measurement. Also there is no information about the size of strain gauges and the way they are attached to the blade as well as information about the material of the blade

  • According to results in the Figure 5 it is not clear what is the reason of such strain distribution on bending of clamped blade. It seem that the closer to the clamping location the gauge is the higher strain it should be, which is not observed on the provided results.

  • Very scarce information is given about the results in Figure 8 and 9. What where the test condition. Where this data collected during flight or laboratory tests? Why normalized length of the blade is from -1 to 1 if previously only 5 strain gauges on the rotor blade where described with location starting from 0? How strain measurements where synchronized with azimuth position of the blade.

  • Is it really necessary to use polynomial of 8th degree to fit such data as in Figure 9

  • What is the meaning of Fourier fitting provided in Figure 13? What is it needed for?

  • It is no explanation how full-field moment distribution in Figure 14 was achieved.

    Overall authors claim that the main novelty of the paper comes from actual in-flight measurements but provide almost no information about in-flight testing, flight conditions, used helicopter of rotor blade and other useful information. The same could be said about given information on applied methods for obtaining results.

    For this work to be beneficial for scientific community in my opinion it should be more open on the used tests and methods.

Author Response

 Firstly, we appreciate your helpful comments and suggestions. We have modified the manuscript carefully. the attachment is the list of our itemized answers and changes.

Reviewer 3 Report

Comments and Suggestions for Authors

With the foreseen popularity of electric aircraft, more research on propellers is necessary. This work reported an application using strain data collected by strain gauges and use it to estimate and analyze the aerodynamic loads on the helicopter blades. The testing is well designed and explained. The conclusion is not innovative but supported well by the testing data. There are a few questions/comments that the authors might consider to make the manuscript better.

1) The introduction elaborated on the development of aerodynamic load models but did cover the available strain sensing methods to collect the data. More specifically, why strain gauges are used rather other techniques, such as Digital Image Correlation (DIC). What are the pros and cons using strain gauges? What would be authors insights on the innovative wireless/non-contact strain measurement methods, such as carbon nanotube sensors (https://www.nature.com/articles/s41598-022-15332-1). It is suggested to discuss the benefits of new strain sensors and their potentials on this topic and cite literatures accordingly.

2) Figure 2 (a) contains non-English letters.

3) Details of the strain gauges used are unclear. Table-1 provides the part number, which is hard to know what it is. 

4) Strain gauges have wires connecting to the data acquisition system. Don't those wires interfere the high speed rotation of blades? Did the authors use a wireless data acquisition system? 

5) It is suggested to show a photo of the blades with strain gauges attached and the data acquisition system.

6) In figure 5, the unit of strain is confused. It is suggested to simply use % ,mε or µε.

7) The title of the table below Figure 5 is missed.

8) Figure 11 is not necessary if the frequency is so high or find a better way to display it. If it's steady, just showing a couple of period would be enough.

9) The units of Figure 14 and 16 are missed. Are they the load map at a specific time frame? Please explain.

10) Explain more on the asymmetric property would be helpful.

Author Response

(The authors gave the same response as above.)
